# Readout Integrated Circuit for Small-Sized and Low-Power Gas Sensor Based on HEMT Device

**DOI:** 10.3390/s21165637

**Published:** 2021-08-21

**Authors:** Seungjun Lee, Joohwan Jin, Jihyun Baek, Juyong Lee, Hyungil Chae

**Affiliations:** Department of Electrical and Electronic Engineering, Konkuk University, Seoul 05029, Korea; lsj170203@konkuk.ac.kr (S.L.); ajddjdj789@konkuk.ac.kr (J.J.); wlgusdl100@konkuk.ac.kr (J.B.); yoeung131@konkuk.ac.kr (J.L.)

**Keywords:** gas sensor, HEMT device, readout integrated circuit, negative charge pump, transimpedance amplifier, delta-sigma ADC

## Abstract

This paper presents a small-sized, low-power gas sensor system combining a high-electron-mobility transistor (HEMT) device and readout integrated circuit (ROIC). Using a semiconductor-based HEMT as a gas-sensing device, it is possible to secure high sensitivity, reduced complexity, low power, and small size of the ROIC sensor system. Unlike existing gas sensors comprising only HEMT elements, the proposed sensor system has both an ROIC and a digital controller and can control sensor operation through a simple calibration process with digital signal processing while maintaining constant performance despite variations. The ROIC mainly consists of a transimpedance amplifier (TIA), a negative-voltage generator, and an analog-to-digital converter (ADC) and is designed to match a minimum target detection unit of 1 ppm for hydrogen. The prototype ROIC for the HEMT presented herein was implemented in a 0.18 µm complementary metal–oxide–semiconductor (CMOS) process. The total measured power consumption and detection unit of the proposed ROIC for hydrogen gas were 3.1 mW and 2.6 ppm, respectively.

## 1. Introduction

Recently, owing to rapid developments in industry and technology, interest in mitigating the severity of air pollution and air quality is increasing. Various harmful gases have been generated that cause air pollution and fatal effects on human health. Damage caused by exposure to such harmful gases is increasing daily. Therefore, it is necessary to set specific standards for air pollutants and their systematic management. Accordingly, the demand for continuously monitoring and detecting harmful components in the air is rapidly increasing. Thus, various gas sensors are currently available on the market. Commercially available gas sensors are classified according to the elements constituting the sensor and the operation method, and each type has its pros and cons. For example, an electrochemical sensor containing a chemical, which detects the current generated by the redox reaction caused by a certain gas, has the advantage of high accuracy and linearity, and it has been continuously developed until recently [1]. The state-of-the-art electrochemical sensor guarantees high sensitivity and good power efficiency, but it is not suitable for domestic use because of its short lifespan and interference from temperature and other kinds of gas [2]. Gas sensors based on high-electron-mobility transistor (HEMT) devices are proposed herein, and their existing shortcomings are improved. HEMT devices have several advantages for use as gas sensors. First, they have high electrical conductivity owing to device characteristics. They respond quickly and sensitively to gas detection. Because of this high sensitivity, they can detect even low gas concentrations effectively. Second, since HEMTs operate at ambient temperatures, separate heating systems are not required, which is beneficial in terms of sensor size and power consumption. Since there is no high-temperature requirement, less time is needed to prepare the sensor for operation. Third, since HEMTs are themselves transistors, gas sensors based on them have small sizes. Furthermore, multi-gas detection is available, which is advantageous for commercialization. Owing to the small size of the sensor, it is possible to detect multiple types of gases by arraying the sensors. Lastly, HEMTs can be used in various systems as ultrasmall, low-power sensors by integrating Internet of things (IoT) functionality. Sensors using HEMTs are actively being developed. However, most applications use HEMTs as single devices [3,4,5]. HEMTs also have a fatal disadvantage in that they are vulnerable to process voltage temperature (PVT) variations. This causes deviations in sensor performance and lowers accuracy, making commercialization difficult. Existing commercial gas sensors have resolved some of the previous disadvantages by combining various sensor, signal processing, and communication modules. However, the size and weight of the sensors themselves increased. If the sensing system can be configured through readout integrated circuits (ROICs) based on complementary metal–oxide–semiconductors (CMOSs), which were previously unavailable, the existing problems may be resolved, and size and power consumption can be improved. In this paper, we propose a low-power, ultrasmall, and high-sensitivity gas sensor by combining HEMT devices with a novel ROIC.

## 2. Proposed System Architecture

### 2.1. High-Electron-Mobility Transistor Device

The sensitivity of a sensing device placed on the front end of a sensor system critically affects not only the performance but also the design of the associated ROIC. The lower the sensitivity of the sensing device is, the more vulnerable the input signal to noise during signal processing will be, and the detection range of the sensor decreases. Therefore, it is necessary to suppress the input referred noise of the ROIC to detect gas concentrations over a wide range, which increases the design complexity of the circuit and size of the system. However, if the sensitivity of the sensing device is high, it is possible to obtain a high signal-to-noise-and-distortion ratio (SNDR) sensor output signal with a simple ROIC design without considering noise since the ROIC has a large input signal. Therefore, it is important to build a sensing device with high sensitivity. Since semiconductor-based HEMTs can easily achieve high sensitivity with small sizes, they are widely used in sensor systems [6,7,8]. HEMTs comprise a heterojunction structure, with wide and narrow bandgap layers. Devices with such structures have high mobilities owing to differences in bandgap between the two layers. The high mobility can be utilized to convert small electrical signals generated at the gate to large current changes. Owing to these characteristics, HEMTs with gates made of materials that generate electrical signals by reacting with specific gas molecules can be used as gas sensing devices with high sensitivity. When the HEMT device used for gas sensing is exposed to a target gas, the gas molecules are separated into ions at the gate and move inside the metal. Thus, gas ions diffuse into the gate metal, applying a positive field to the gate and a negative field to the body surface to cause changes in the threshold voltage ($V_{\mathrm{th}}$). If the gate, drain, and source terminals of the HEMTs are biased with constant voltages, changes in $V_{\mathrm{th}}$ cause changes in the drain current because the amount of change in $V_{\mathrm{th}}$ differs on the basis of the gas concentration at the gate. By measuring the amount drain current change, the concentration of the gas exposed to the gate can be estimated.

Figure 1a shows an example of the shift in the I-V curve as a function of changes in the concentration of hydrogen applied to the HEMT sensor. A higher hydrogen concentration resulted in a more negative value of $V_{\mathrm{th}}$, a greater shift of the I-V curve to the left, and a greater drain current flow for a constant bias voltage. The sensitivity is defined by Equation (1) [9,10,11] and may vary depending on the gas concentration since it is determined according to the amount of current change before and after target gas detection. Therefore, it is necessary to maximize the sensitivity according to the concentration range of the gas of interest. A greater amount of current change before and after exposure to the gas leads to a higher sensitivity.
(1)$$\mathrm{Sensitivity}=\frac{I_{\mathrm{GAS}}{\;-\;I}_{\mathrm{AIR}}}{I_{\mathrm{AIR}}}.$$

In Equation (1), $I_{\mathrm{AIR}}$ is the current flowing in the drain of the HEMT when exposed to a gas mixture of N_2_ and O_2_. $I_{\mathrm{GAS}}$ is the current flowing when exposed to N_2_ and O_2_ along with H_2_. Data1 and data2 of Figure 1b are the sensitivity plots obtained by changing the gate voltage for 50 ppm and 100 ppm of hydrogen. As the concentration of hydrogen increases, the sensitivity also increases. Even when exposed to the same concentration of hydrogen, it is seen that the sensitivity changes because the amount of increase in current varies according to the bias voltage. Therefore, a high-accuracy gas sensor can be manufactured only when the gate is biased for the greatest current change in the gas concentration range. Comparing Figure 1a,b, the points corresponding to $V_{\mathrm{th}}$ and peak sensitivity are similar but not identical. However, since there is a correlation between the two values, it is important to improve the sensitivity to determine the exact $V_{\mathrm{th}}$ of the device.

Although sensitivity is determined by the gate voltage bias, its maximum value is limited by factors such as the material and geometry used for the gate during device fabrication. In this study, a gas sensor system for hydrogen detection is constructed using the HEMT device manufactured according to [12] to maximize sensitivity. The fabricated HEMT device has a body composed of AlGaN/GaN, as shown in Figure 2, and a two-dimensional electron gas (2DEG) layer [13,14] with high carrier concentration is generated in the body. The gate uses both Pt and graphene, which not only renders the manufacturing process easier but also has the advantage of increasing sensitivity through a high surface-to-volume ratio. Therefore, it is possible to operate the sensor at ambient temperatures without a separate heating system and with low power consumption. In addition, leakage currents can be reduced by adding HfO_2_, an insulator, between the gate and body, to increase sensitivity by reducing the off current. Thus, HEMT devices configured in this way can be used as gas sensors with advantages of low power, small size, and high sensitivity.

Since the HEMT in Figure 2 is in depletion mode, it has a negative value for $V_{\mathrm{th}}$. In principle, this negative $V_{\mathrm{th}}$ changes only according to the hydrogen concentration, but it may also be considerably affected by PVT variations outside of gas detection. If the HEMT devices have different $V_{\mathrm{th}}$ values owing to variations, the accuracy of gas detection is greatly reduced because the current changes for a given gas concentration are different. When two HEMT device samples are measured in the same environment, different results may be obtained. To confirm the effect of only PVT variation, two samples were measured in air condition. As shown in Figure 3a, when a fixed voltage of 1 V was applied to the drain, a different I-V plot was obtained, as shown in Figure 3b. These samples had a $V_{\mathrm{th}}$ of −5.7 V and −6.0 V, indicating that the accuracy of the gas sensor was greatly affected by PVT variation because the change in $V_{\mathrm{th}}$ for PVT variation was larger than the change in $V_{\mathrm{th}}$ for gas detection. Since the I-V and sensitivity plots were shifted by differences in $V_{\mathrm{th}}$, if the actual $V_{\mathrm{th}}$ can be found for each device, exact gas concentrations can be determined through digital signal processing.

If the I-V curve is obtained by sweeping the voltage applied to the gate of a given HEMT, the difference in $V_{\mathrm{th}}$ caused only by PVT variation can be identified by comparison with the reference device. Thereafter, $V_{\mathrm{th}}$ changes due to gas concentrations can be determined so that the accuracy of gas sensing can be improved. If a gate voltage that maximizes sensitivity of the HEMT is applied on the basis of the measured $V_{\mathrm{th}}$, the sensor can be operated optimally regardless of variation. The next section introduces the configuration and operation of the ROIC with this functionality.

### 2.2. Proposed Readout Integrated Circuit and Calibration

Since the $V_{\mathrm{th}}$ of the HEMT device is affected by PVT variations and has a negative value, a block capable of supplying a wide range of negative voltages is required for optimal gate biasing for high sensitivity. After biasing the gate of the HEMT through the negative voltage supplier, a block measuring the current flowing to the drain is needed. A transimpedance amplifier (TIA) that can sense changes in the current and transmit a signal in the form of a voltage to the subsequent blocks is, therefore, required. In the absence of a sensing gas, a very small drain current flows because the HEMT device is close to the off state. However, when exposed to gas, the amount of current varies greatly depending on concentration. Therefore, it is necessary to measure current in a wide range, and variable gain is required for the TIA. The TIA is the first block of signal processing, and, since the performance of the TIA dominates the overall signal-to-noise-and-distortion ratio (SNDR), it is important to attenuate the input referred noise. In particular, since the gas concentration corresponds to a low-frequency signal, it is vulnerable to TIA offsets and flicker noise, which must be attenuated. The signal is converted into voltage by the TIA and amplified before conversion to a digital signal through an analog-to-digital converter (ADC). The ADC requirements are determined by the sensitivity of the manufactured HEMT device and gain of the TIA. When detecting a concentration of at least 1 ppm through the HEMT device, a current change of 10 nA occurs, and the minimum signal amplitude is about 2 mV according to the voltage converted by the TIA with a maximum resistance of 200 kΩ. Considering that the ADC’s differential full-scale range (FSR) is 3.3 V, the minimum effective number of bits (ENOB) required for the ADC is 11 bits. To achieve more than 11 bit ENOB, a high resolution of 13 bits or more must be used in consideration of noise and distortion factors, and a suitable ADC structure should be adopted.

Figure 4 shows the ROIC, including the negative voltage supplier, TIA, and ADC. Through digital signal processing of the output signal from the ADC, it is possible to detect the gas concentration. The ROIC must be able to detect and correct PVT variations of the HEMT device to allow accurate gas sensor manufacturing. Hence, the identification of $V_{\mathrm{th}}$ is prioritized. The ROIC needs a digital controller to accurately find $V_{\mathrm{th}}$ that changes in the negative voltage range and to calibrate the basic blocks accordingly. The process of calibration using the digital controller is shown in Figure 5.

To correct the PVT variations affecting the $V_{\mathrm{th}}$ of the HEMT device, characteristic data for a reference sample and a look-up table showing the characteristic changes according to changes in gas concentration are required. If information on the reference is obtained through premeasurements, the shift in $V_{\mathrm{th}}$ can be obtained by sweeping the negative voltage for the I-V curve in the full range of the HEMT. By shifting the existing sensitivity plot similarly, the sensitivity is calculated, and the section with high sensitivity is identified. Thereafter, the exact concentration of the detected gas can be estimated by connecting the ROIC to the new HEMT device and comparing the results obtained with the look-up table. By applying the software and digital controller for calibration to the proposed gas sensor system, gas sensing is possible with high accuracy without a complicated process, regardless of variation.

## 3. Circuit Implementation

### 3.1. Negative Voltage Generator

A block capable of generating a wide range of negative voltages is required for the gate bias when considering $V_{\mathrm{th}}$ that changes with PVT variation and gas detection. The negative voltage supplier presented here is shown in Figure 6 and comprises a negative charge pump (NCP) core, a low-dropout (LDO) regulator, an inverter chain, and a resistive digital-to-analog converter (DAC) to control the reference voltage of the LDO regulator.

Because of its simplicity, the NCP core has a conventional doubler-based NCP structure, as shown in Figure 7a [15]. From actual measurements of the HEMT device, a wide voltage range of −8 V to −0.5 V is required. When a supply voltage of 3.3 V is used, a single NCP core outputs in the range of −4 V to −0.5 V. Therefore, two cores were used in a cascade structure to meet the required conditions.

The NCP core consists of a pumping capacitor ($C_{P}$), a metal–oxide–semiconductor field-effect transistor (MOSFET) switch, and a load capacitor ($C_{L}$). Owing to CLK and CLKB, charges accumulate in the pumping capacitor and are transferred to the load capacitor by switching, resulting in a negative voltage. The MOSFETs corresponding to the switch continuously transfer charge to the load capacitor and consequently generate an output voltage lower than −VDD. A high-voltage device was used to withstand the resulting voltage stress. In the case of a PMOS switch, since the voltages at all terminals are negative, even if the N-well is grounded, a latch-up problem due to forward bias does not occur. In the case of an NMOS switch, the body should be biased to the lowest voltage, and, since such a voltage cannot be supplied separately, a triple-well is used to separate the body from the substrate and connect it to the source terminal. If charge is pumped through continuous switching, spikes occur, which cause momentary forward bias toward the body, resulting in loss of charge and a reduction in the range of achievable negative voltages. To prevent this, a resistor is used to connect the body of the switch to minimize leakage of electric charge. In this structure, it is the high-level voltage of CLK and CLKB that determines the output voltage range. Since the main clock from the outside passes through the inverter chain, the supply voltage of the inverter chain eventually determines the output voltage of the NCP. The LDO regulator is used to digitally control this on-chip VDD, as shown in Figure 7b, along with a resistive DAC applied to the reference voltage of the LDO regulator. In addition, the bias current of the resistive DAC is designed to be digitally controlled so that VDD can be finely controlled. The power of the LDO regulator is also digitally controlled. With this on-chip control function, a negative voltage that maximizes sensitivity is applied to the gate regardless of variations in the HEMT device. Figure 8 shows the simulated transient response of the designed NCP. It is seen that, when one core is used, the output voltage is −4 V, and, when two cores are used, the negative output voltage greater than −8 V. The period of the clock signal used for the NCP is 500 ns, and the glitches caused by this have little effect on operation.

### 3.2. Transimpedance Amplifier with Nested Chopper

Figure 9 shows the TIA composed of a main amplifier, nested chopper, digital controller, and clock generator.

The main amplifier that magnifies the voltage consists of two stages. The first stage has a complementary differential amplifier structure [16], as shown in Figure 10a. As shown in Figure 10b, the conventional differential amplifier has a relatively simple structure that is easy to design, with a wide output swing range. However, assuming the same power consumption and bandwidth, the gain of this stage is smaller than that of the complementary differential amplifier. The input MOSFETs of the complementary differential amplifier are designed to operate in the weak inversion region [17,18,19]. Since the value of $g_{m}/i_{D}$ in the weak inversion region is larger than that in other regions, it is advantageous to achieve a higher gain. In addition, power consumption can be minimized when the amplifier operates in a low-frequency band. Since both NMOS and PMOS transconductance ($g_{m}$) are used, thermal noise can be suppressed, which is advantageous for sensing.

The second stage of the main amplifier consists of a combination of the folded structure for voltage amplification and a class-AB buffer, as shown in Figure 11. The second stage was designed focusing on wide voltage swing and low static power consumption. By placing the class-AB buffer [20,21,22] at the output stage, it is possible to achieve a relatively wide voltage swing through push–pull operation while reducing static current. Despite these advantages, the number of nodes increases owing to the combination of the two structures. This increases the number of poles and zeros, causing stability problems. To solve this problem, feedback resistors and capacitors were added to the input and output terminals of the second stage.

Since the HEMT device has a very large current change range during gas detection, a TIA gain control is required to measure current in a wide range. In addition, since the drain voltage may have to be changed according to various operating environment changes of the device, it is necessary to control the input common-mode voltage. Thus, several parameters can be controlled inside the chip, as shown in Figure 12. A total of four sets of feedback resistors and switches were configured for gain control, and the common-mode voltage was also adjusted in the four levels through resistive DACs. Because the stability of the second stage changes according to the common-mode voltage, the feedback resistor and capacitor were also added and controlled. Through digital control, it is possible to detect changes in HEMT operation in various environments.

As noted earlier, the input referred noise of the TIA dominates the overall SNDR. In particular, flicker noise suppression is important for operation in the low-frequency band, and offset removal is also important. The red plot in Figure 13 shows the simulated noise distribution in the TIA when there is no chopper, and the total amount of noise is 5.195 µA/√Hz. To suppress this, as shown in Figure 14, the nested chopper technique [23,24,25] using two high-frequency and two low-frequency choppers was applied to the input and output terminals of the amplifier. Although flicker noise can be removed through a sequential process of signal modulation, amplification, and demodulation unlike the conventional chopper [26], which has a critical problem where the offset remains in the chopping frequency, the nested chopper can remove and attenuate offsets with the outer pair of choppers. However, the input frequency is limited to $f_{\mathrm{low}}/2$ but is not a problem because the operating frequency of the proposed sensor is less than $f_{\mathrm{low}}/2$. Frequencies of 5 kHz and 20 Hz were used for $f_{\mathrm{low}}$ and $f_{\mathrm{high}}$, respectively, and the offset was reduced (theoretically) by more than 250 times due to the ratio $f_{\mathrm{low}}{/f}_{\mathrm{high}}$ while attenuating flicker noise with the nested chopping technique. This effectively suppressed the flicker noise and dominant offset in the system. Thus, the input referred noise shown in the blue plot of Figure 13 was achieved, and the total noise was 613.7 nA/√Hz.

Figure 15 shows the transient response under the maximum output swing condition of the designed TIA. A single input current was applied only to one end of the TIA, with a nominal common-mode voltage of 1 V and a 5 kΩ feedback resistor.

### 3.3. Third-Order Delta-Sigma ADC

The signal converted into voltage and amplified by the TIA must be converted to a digital signal through the ADC to determine gas concentration through digital signal processing. In this work, the ADC shown in Figure 16 was used, and its block diagram is shown in Figure 17.
(2)$$\frac{{\left({1\;-\;z}^{-1}\right)}^{3}}{z^{3}\;-\;2.45z^{2}\;+\;2.038z\;-\;0.5531}.$$

The ADC is a third-order delta-sigma modulator introduced in [27] and consists of an integrator, a multi-input comparator [28,29], a 1 bit quantizer with lower design complexity than the multibit quantizer, and a digital controller. As mentioned in Section 2, the required ADC minimum ENOB is 11 bits, and the target ADC resolution for this is 13 bits. The ADC in Figure 16 is sufficient for achieving high resolution in the low-frequency band, which is the operating frequency band of the gas sensor. The gain of each integrator is equal to ¼, as shown in Figure 17, which allows the input transistor size to remain constant in the design of the multi-input comparator for signal summation in each path. This helps prevent performance degradation. The noise transfer function (NTF) obtained from Figure 17 is as given in Equation (2), and the corresponding pole–zero plot is shown in Figure 18. Since all poles and zeros are located within the unit circle, the ADC stability is guaranteed.

Although the structure presented here is similar to that in [27], the ADC in this study was designed to further minimize power consumption because it has a low BW of 100 kHz. Since the first integrator dominates the overall ADC performance, the power consumption ratios of the three integrators are scaled as 1:0.6:0.5 to significantly lower the overall power consumption. To this end, the pseudo-differential amplifier [30,31,32] and feedback capacitor used in the integrator were designed separately for each power budget. As shown in Figure 19, the pseudo-differential amplifier consists of an inverter-based amplifier stage and an adaptive LDO regulator [33]. The adaptive LDO regulator prevents overcurrent caused by absence of the tail current in the amplification stage and reduces the effects of PVT variation. By setting the ratio of the inverter-based amplifier device to LDO regulator replica device as 4:1, the LDO regulator can supply VDD stably while minimizing power consumption. In addition, it controls the VDD of the pseudo-differential amplifier by adjusting the amount of current used. Through optimization, a high SNDR of 92 dB and low power consumption of 28 µW were achieved in simulations. The power spectral density through post-layout simulation of the implemented ADC is shown in Figure 20 and has a slope of 60 dB/dec by applying the third-order noise shaping technique. Hence, a 15 bit ENOB was obtained, resulting in more than the target ENOB, through which high-quality ROIC outputs could be obtained.

The ADC, negative voltage supplier, TIA, and HEMT device introduced in the previous section constitute the overall system, as shown in Figure 21. Stable and accurate gas detection is possible with this system by calibration through a digital controller.

## 4. Measurements

The ROIC for the gas sensor based on the HEMT device system proposed in this paper was implemented in a 0.18 µm CMOS process. The photographs of the dies of the ROIC and detailed negative voltage generator, TIA, and ADC are shown in Figure 22 and Figure 23, respectively. The total area of the ROIC, including the digital controller occupying an area of 150 µm × 100 µm, is 2055 µm × 1275 µm. The printed circuit board (PCB) for testing the prototype system was configured as shown in Figure 24a and has a size of 49.7 mm × 49.7 mm. It was tested in a hydrogen gas chamber, as shown in Figure 24b. The test board sends and receives signals to and from a PC through an external FPGA board.

Figure 25a shows the I–V curve obtained by measuring the drain current after applying a negative gate voltage to the four fabricated HEMT devices. It is seen that each device has a different $V_{\mathrm{th}}$ and gain owing to process variations. Figure 25b shows the final digital values obtained by measuring the drain currents using the proposed ROIC prototype. An external 200 kHz clock and 8 µA bias current were applied to the negative voltage generator, and the output voltage was changed from −0.5 to −8.5 V by controlling the LDO regulator power and resistive DAC with a digital controller. In addition, to obtain the overall I-V curves for a wide range of measurements, the feedback resistance of the TIA was set to 5 kΩ and not the maximum value. As shown in Figure 25c, if the digital-value-based plot of Figure 25b obtained through the ROIC is scaled, it is almost identical to the plot of Figure 25a. Table 1 shows the measured results of the gas sensor prototype for changing hydrogen concentrations. The error range for multiple measurements for the same concentration was 2.6 ppm, which is higher than the target resolution of 1 ppm. The cause of such errors is presumed to be a slight change in the gas concentration caused by the unclosed nature of the gas chamber. Furthermore, the input referred noise of the TIA and ADC may have been higher than the values predicted through simulation.

## 5. Conclusions

In conclusion, this paper proposed a gas sensor that combines an AlGaN/GaN-based HEMT device with an ROIC. Unlike conventional HEMT sensors, it is possible to detect gases of interest with consistent performance through simple calibrations via digital signal processing using the ROIC. Although the accuracy was slightly lower than expected (2.6 ppm), it is seen that the proposed gas sensor system operated successfully in detecting gas changes and concentrations accurately despite process variations through digital signal processing. This ROIC consumes 3.1 mW of power and has a size of 2.62 mm^2^. Thus, it is a gas sensor with low power, small size, and high sensitivity that can be used for IoT applications in various fields.

## Figures and Tables

**Figure 1 sensors-21-05637-f001:**
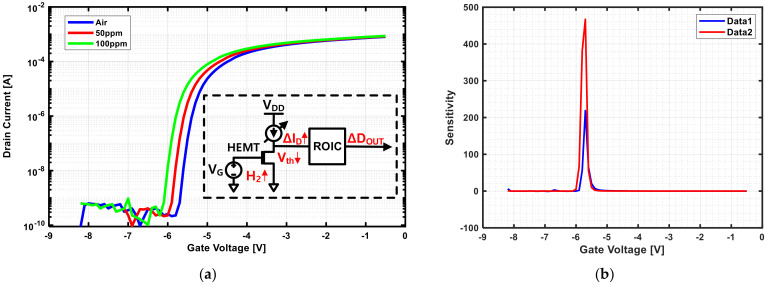
(**a**) HEMT device I-V curve and circuit parameters varying by gas (H_2_) concentration; (**b**) sensitivity plot.

**Figure 2 sensors-21-05637-f002:**
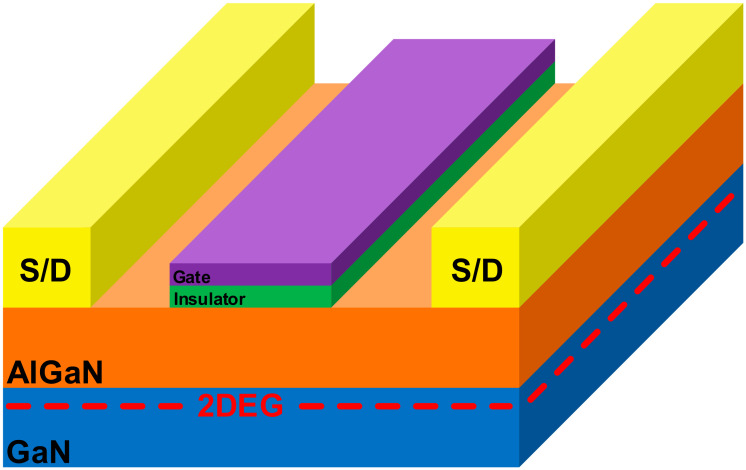
HEMT device structure.

**Figure 3 sensors-21-05637-f003:**
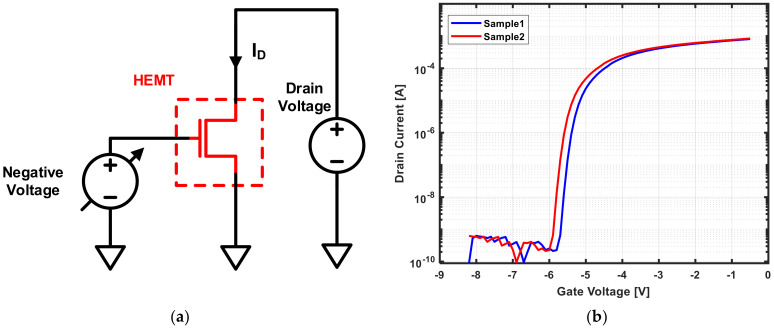
(**a**) HEMT device diagram; (**b**) HEMT device I-V curve.

**Figure 4 sensors-21-05637-f004:**
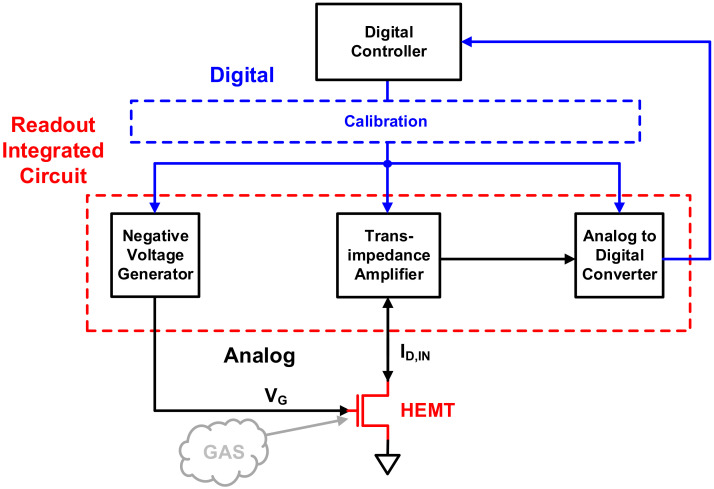
Proposed readout integrated circuit system.

**Figure 5 sensors-21-05637-f005:**
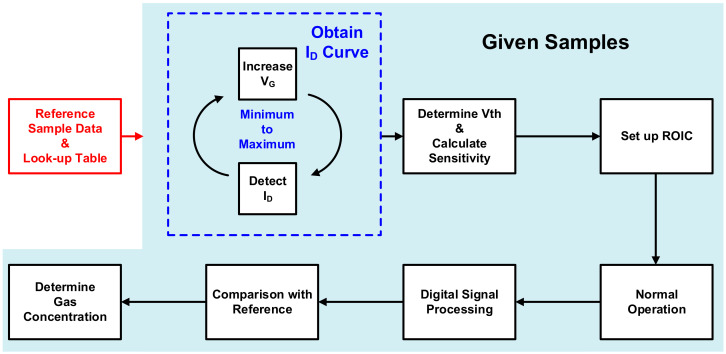
Calibration flow.

**Figure 6 sensors-21-05637-f006:**
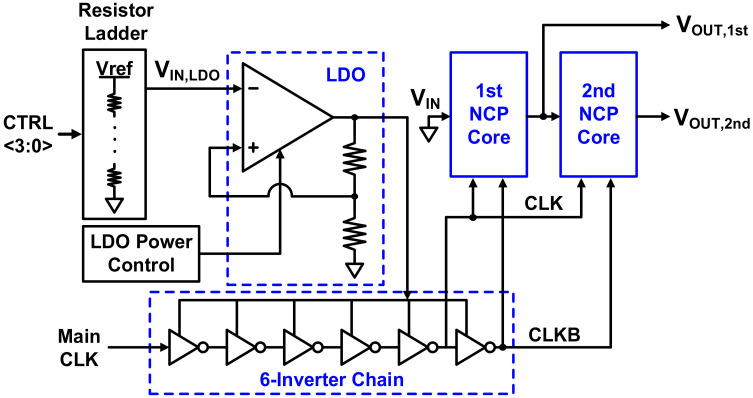
Negative voltage generator.

**Figure 7 sensors-21-05637-f007:**
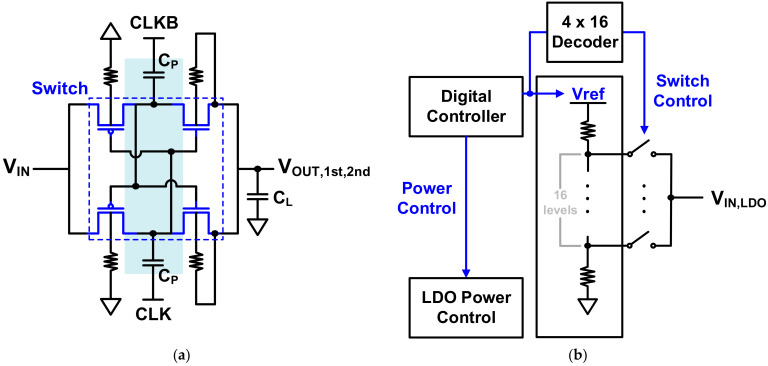
(**a**) NCP core (conventional doubler-based negative charge pump); (**b**) negative voltage generator digital control for calibration.

**Figure 8 sensors-21-05637-f008:**
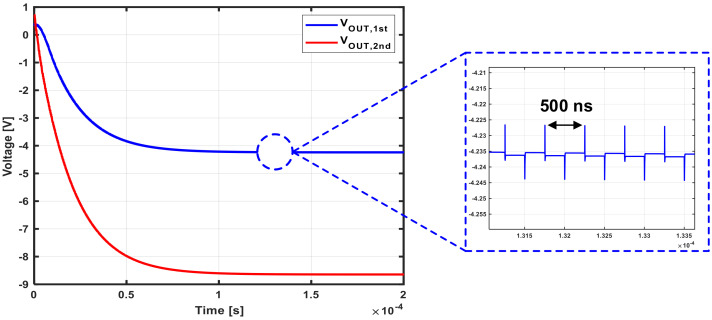
Simulated negative voltage generator outputs.

**Figure 9 sensors-21-05637-f009:**
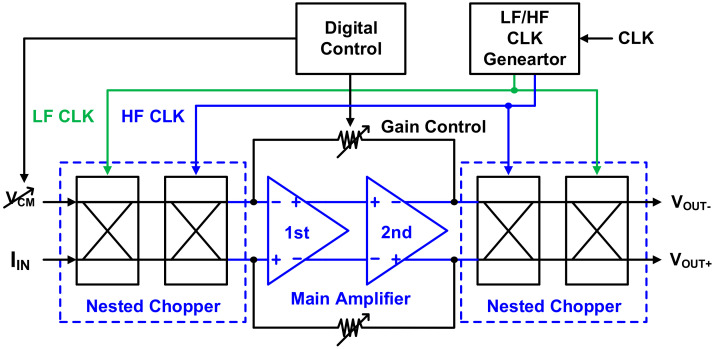
Transimpedance amplifier with nested chopper.

**Figure 10 sensors-21-05637-f010:**
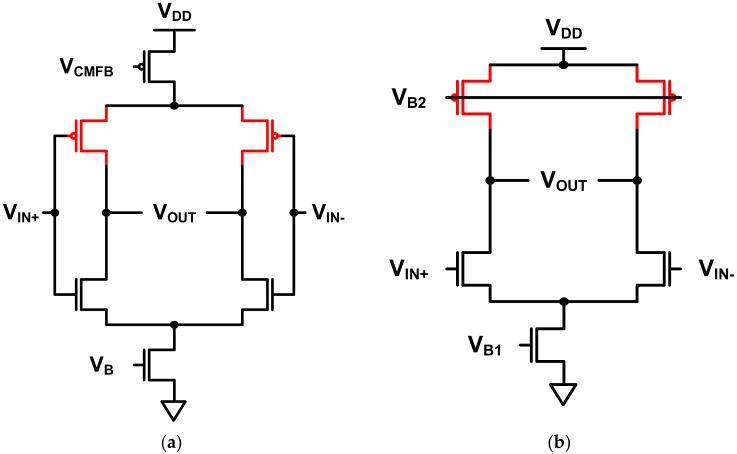
(**a**) Complementary differential amplifier; (**b**) conventional differential amplifier.

**Figure 11 sensors-21-05637-f011:**
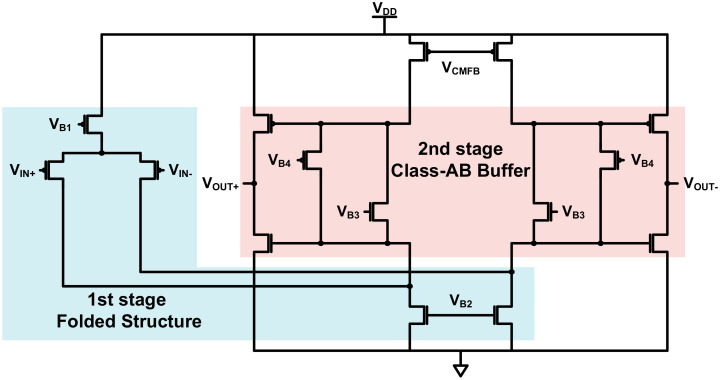
Folded structure and class-AB buffer.

**Figure 12 sensors-21-05637-f012:**
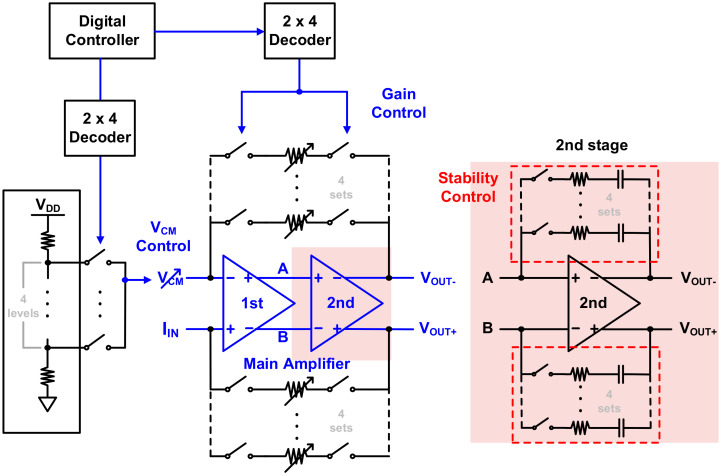
TIA digital control for calibration.

**Figure 13 sensors-21-05637-f013:**
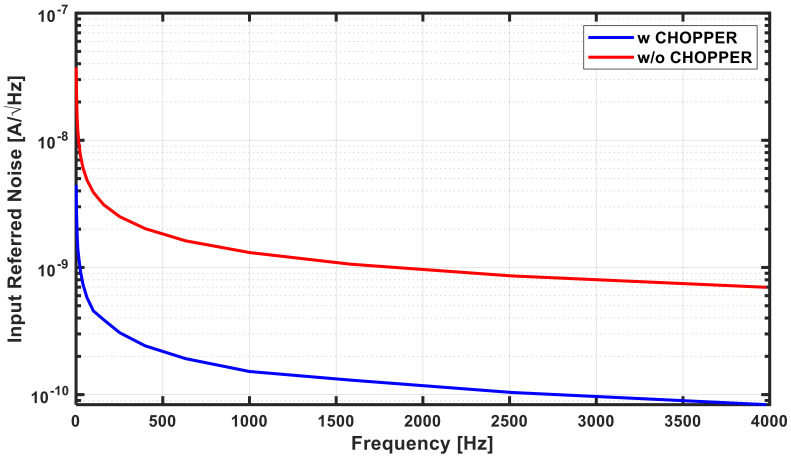
Input referred noise of transimpedance amplifier.

**Figure 14 sensors-21-05637-f014:**
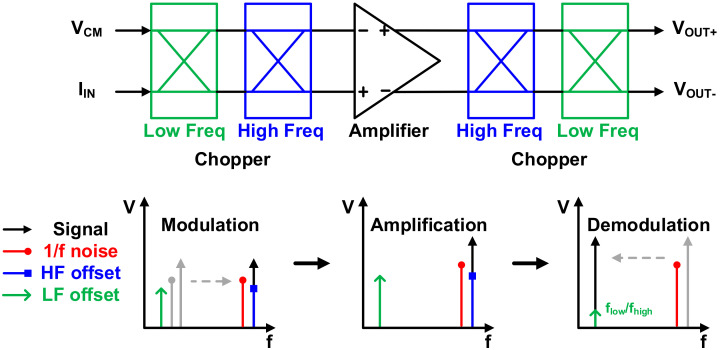
Nested chopper structure and operation.

**Figure 15 sensors-21-05637-f015:**
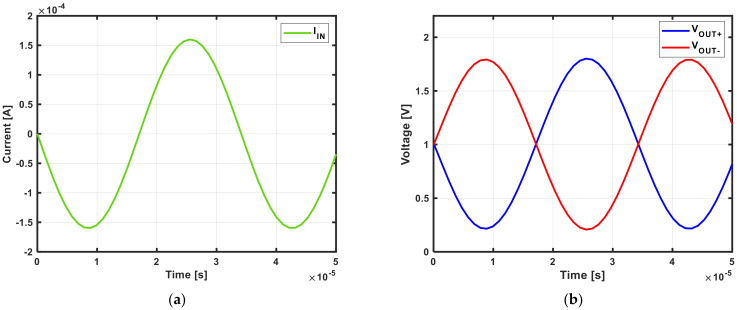
Simulated TIA transient response: (**a**) single-ended input current; (**b**) differential output voltage.

**Figure 16 sensors-21-05637-f016:**
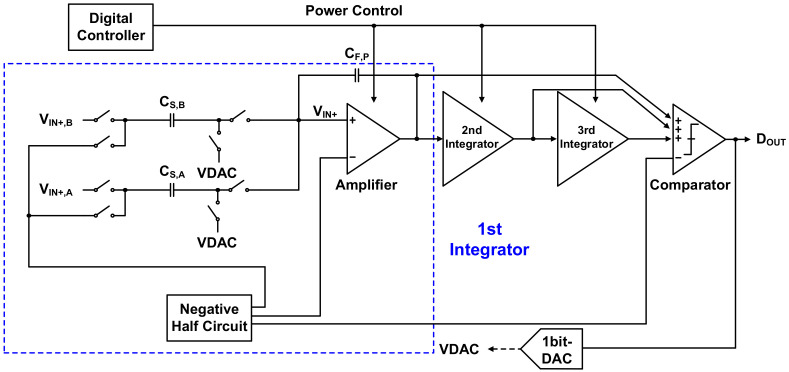
Third-order delta-sigma analog-to-digital converter.

**Figure 17 sensors-21-05637-f017:**
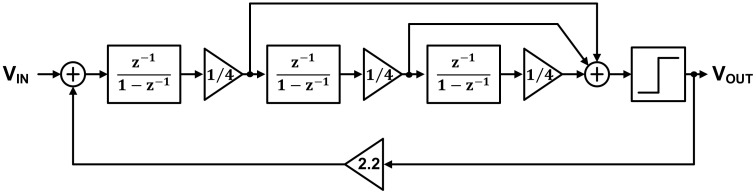
ADC block diagram.

**Figure 18 sensors-21-05637-f018:**
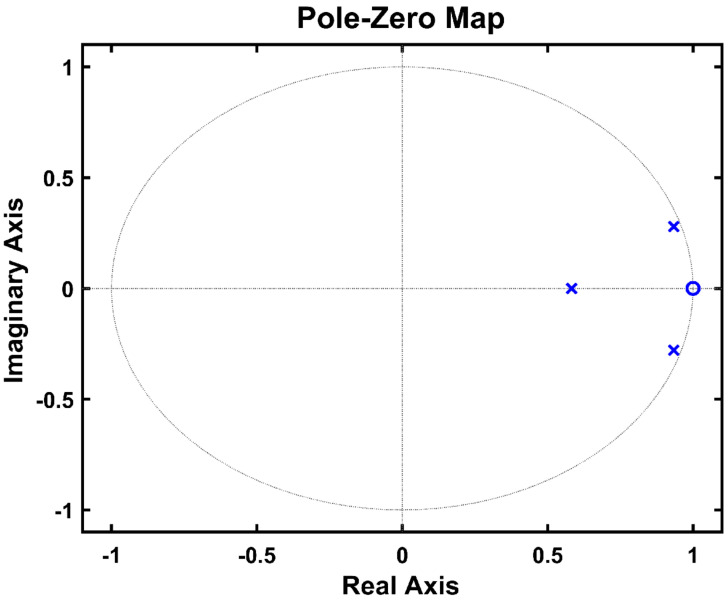
Pole–zero plot.

**Figure 19 sensors-21-05637-f019:**
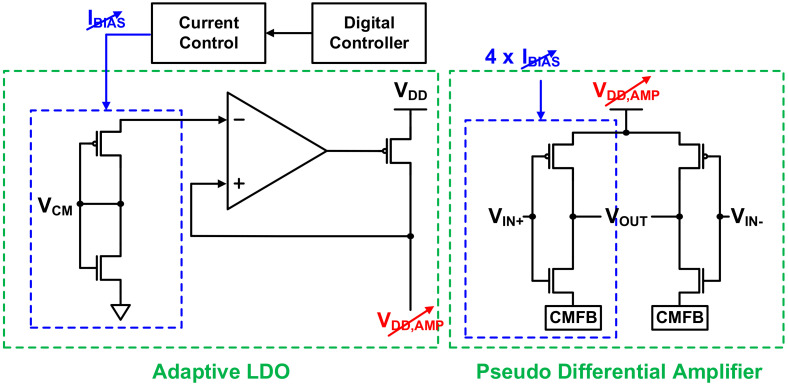
Main amplifier structure consisting of adaptive LDO and pseudo-differential amplifier.

**Figure 20 sensors-21-05637-f020:**
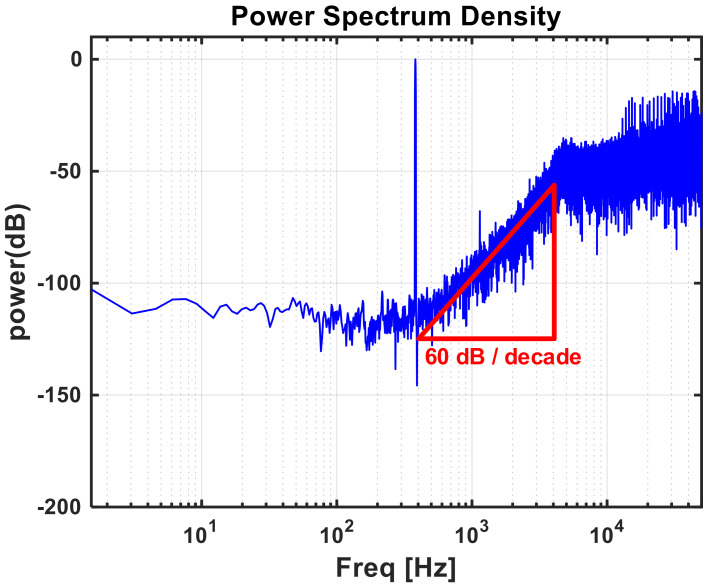
Power spectral density of third-order delta-sigma ADC output.

**Figure 21 sensors-21-05637-f021:**
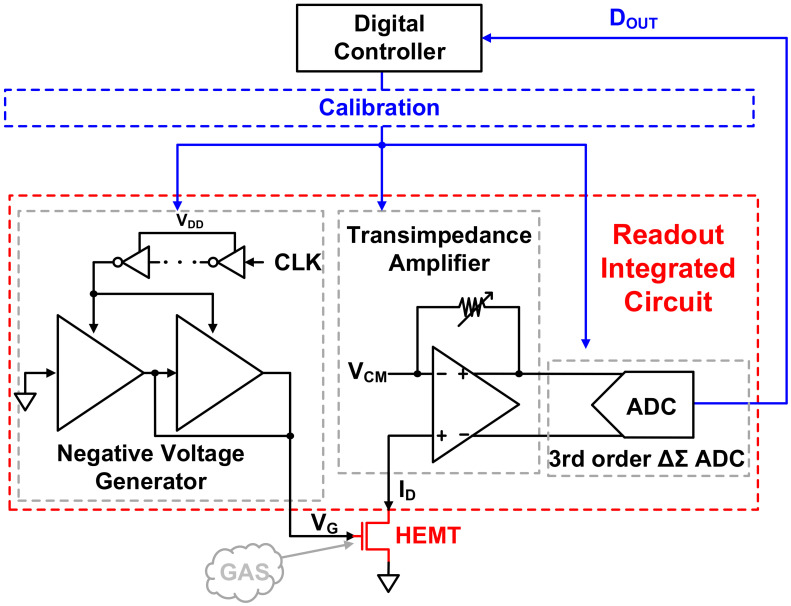
Proposed gas sensing system with ROIC and HEMT device.

**Figure 22 sensors-21-05637-f022:**
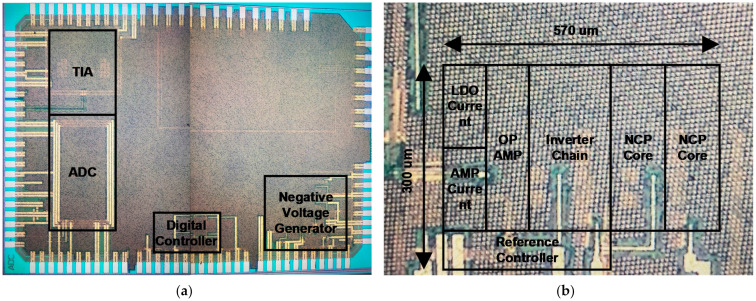
Die photograph of ROIC prototype fabricated in 0.18 um CMOS process: (**a**) top; (**b**) negative voltage generator.

**Figure 23 sensors-21-05637-f023:**
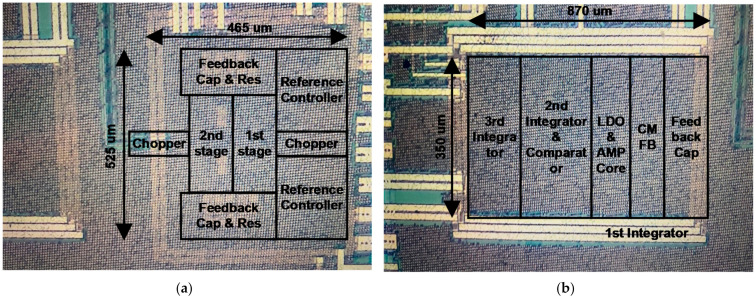
Die photograph of ROIC prototype fabricated in 0.18 um CMOS process: (**a**) TIA; (**b**) ADC.

**Figure 24 sensors-21-05637-f024:**
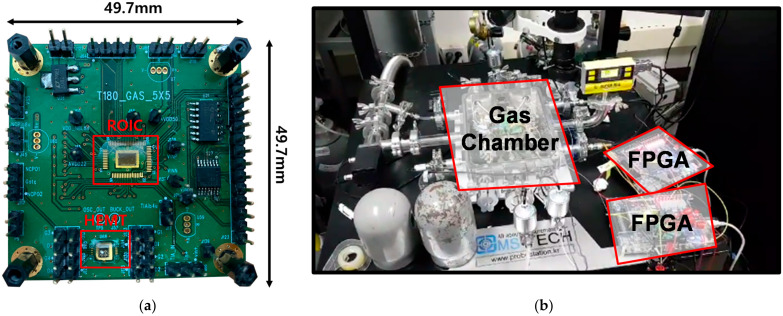
(**a**) Prototype PCB; (**b**) prototype test setup.

**Figure 25 sensors-21-05637-f025:**
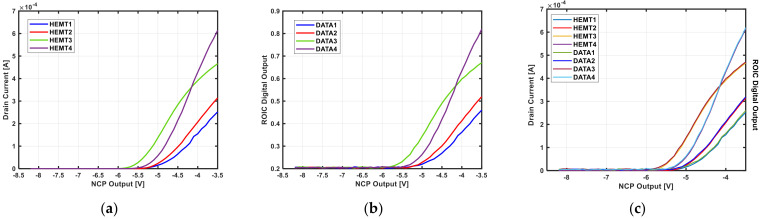
(**a**) HEMT device I-V curve; (**b**) digital output of ROIC; (**c**) comparison of (**a**) and scaled (**b**).

**Table 1 sensors-21-05637-t001:** Measured error as a function of H_2_ concentration.

H_2_ Concentration (ppm)	Result (Digital Code)	Error (ppm)
Air	50	0
50	63	1.2–2.6
100	78	1.8–2
500	99	1.2–1.5
1000	116	1.8–2.1

## Data Availability

Not applicable.

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
