# Peer review of "Readout Integrated Circuit for Small-Sized and Low-Power Gas Sensor Based on HEMT Device"

_sensors, 2021, doi:10.3390/s21165637_

Round 1
Reviewer 1 Report
Overall, this paper is very well-written and presents an interesting approach to measure H2 gas. However, it is far too long and contains too many circuit diagrams. Many of these figures could be moved to an appendix or supplementary information section.
The authors note in the introduction that modern gas sensors contain 'chemicals' which make them unsuitable for domestic use. They should spend 1-2 sentences describing how these sensors work, specifically the state-of-the-art for H2 gas. On the topic of domestic use, does the device presented in this paper require calibration with H2 gas reference? If so, how is this suitable for a domestic setting? The authors should provide more detail about the reference sample required for this calibration, and discuss how this device would ultimately improve upon state-of-the-art H2 gas sensors. This discussion should include a comparison of the measurement error between several commercial devices and the device the authors present.
Minor addition to Section 2.1
- SNDR not defined
Author Response
Thank you for your sincere review.
Please see the attachment.
Thank you.

Reviewer 2 Report
- If the authors mention the manufacture of two HMETs, I would expect to see the complete characterization of the devices using a certain range of gas concentrations. There is not shown the results that demonstrate what the authors mention; “The higher the hydrogen concentration, the more negative is the value of VthThe greater the amount of current change before and after exposure to the gas, the higher is the sensitivity obtained….. The greater the amount of current change before and after exposure to the gas, the higher is the sensitivity obtained.” For example, there is not a figure thar show gas concentration as a function of current change, which corresponds to a Vth.
A minor observations:
- In Introduction section, the authors mention that there are many sensors on the market that contain chemicals and are not for domestic use. I suggest mentioning some of these sensors.
- Authors mention that “If the gate, drain, and source terminals of the HEMTs are biased with constant volt-ages, changes in Vth cause changes in the drain current because the amount of change in Vth differs based on the gas concentration at the gate. By measuring the amount drain current change, the concentration of the gas exposed to the gate can be estimated.”, I suggest including in figure 1, a figure of the electrical circuit and indicate the variables that change when the gas concentration changes.
- Under what conditions were the results of figure 3b obtained? The authors do not mention the gas concentration.
Author Response

(The authors gave the same response as above.)

Round 2
Reviewer 2 Report
The author made the corrections in the paper according to my requirements/suggestions.